# Improved Mechanical Properties and Corrosion Resistance of Mg-Based Bulk Metallic Glass Composite by Coating with Zr-Based Metallic Glass Thin Film

**Pei-Hua Tsai [1], Chung-I Lee [1], Sin-Mao Song [1], Yu-Chin Liao [2] , Tsung-Hsiung Li [1], Jason Shian-Ching Jang [1,2,\*] and Jinn P. Chu [3]**

1  Institute of Materials Science and Engineering, National Central University, Taoyuan 32001, Taiwan; peggyphtsai@gmail.com (P.-H.T.); worm30221@gmail.com (C.-I.L.); bear82112760103@gmail.com (S.-M.S.); pshunterbabu@gmail.com (T.-H.L.)
2  Department of Mechanical Engineering, National Central University, Taoyuan 32001, Taiwan; llllurker@gmail.com
3  Department of Materials Science and Engineering, National Taiwan University of Science and Technology, Taipei 10607, Taiwan; jpchu@mail.ntust.edu.tw
\*  Correspondence: jscjang@ncu.edu.tw

**Abstract:** Mg-based bulk metallic glass (BMG) and its composite (BMGC) can be excellent candidates as lightweight structure materials, but lack of anti-corrosion ability may restrict their application. In order to enhance the natural weak point of Mg-based BMGC, a 200-nm thick Zr-based metallic glass thin film (MGTF) $((Zr_{53}Cu_{30}Ni_9Al_8)_{99.5}Si_{0.5})$ was applied and its mechanical properties as well as its corrosion resistance were appraised. The results of a 3-point bending test revealed that the flexural strength of the Mg-based BMGC with 200-nm thick Zr-based MGTF coating can be greatly enhanced from 180 to 254 MPa. We propose that the Zr-based MGTF coating can help to cover any small defects of a substrate surface, provide a protecting layer to prevent stress concentration, and cease crack initiation from the specimen surface during bending tests. Moreover, the results of anti-corrosion behavior analysis revealed a similar trend between the Mg-based BMG, Mg-based BMGC, and Mg-based BMGC with Zr-based MGTF coating in 0.9 wt.% sodium chloride solution. The readings show a positive effect with the Zr-based MGTF coating. Therefore, the 200-nm thick Zr-based MGTF coating is a promising solution to provide protection for both mechanical and anti-corrosion behaviors of Mg-based BMGC and reinforce its capability as structure material in island environments.

**Keywords:** Mg alloy; bulk metallic glass; composites; thin film coating; mechanical properties

## 1. Introduction

Zr-, Ti-, Ni-, and Fe-based alloys bulk metallic glasses (BMGs) have been well studied in the past few decades [1–3]. Ti-based BMGs are an excellent candidate for bio-application due to their relatively low density and much more compatible Young's modulus in comparison with stainless steel or Co-Cr-Mo alloy. Several toxic-element-free Ti-based BMG alloy systems have been developed with good glass-forming ability [4,5]. Ni- and Fe-based BMGs generally possesses excellent mechanical properties and can be promising structure materials [6–8]. In addition, Fe-based BMG exhibits extremely high hardness around 1200 Hv and excellent anti-wear resistance ability, meaning it can be used in medical tool parts and for surgical blades with better durability [9]. Mg-based BMG possesses the

low-density advantage and can be designed as a lightweight component for the automotive, aerospace, and 3C industries [10–13]. Yet, monolithic Mg-based BMGs show a very brittle behavior and will break into pieces before yielding [10,11]. To conquer the problem of brittleness, extensive efforts have been devoted to develop Mg-based metallic glass composites (BMGCs) with homogeneous micro- or nano-scaled second-phase dispersion in a BMG matrix in the past decade. These include the incorporation of in-situ precipitation of micro- or nano-crystalline phases and ex-situ added micro-sized refractory ceramics or ductile metal particles in the Mg-based BMGCs [14–19]. Mg-based metallic glass composite reinforced with Nb particles can reach the high strength of 900 MPa and large plasticity of 12.1% ± 2% [14]. In situ addition of Mg flakes into Mg–Cu–Y–Zn BMG alloy can significantly improve mechanical properties such as compressive plastic strain up to 18% and ultimate strength up to 1.2 GPa [15]. Moreover, ex-situ addition of 40 vol.% Ti spherical powder improved the ductility from 0% (monolithic glass) to 41% plastic deformation for the composite [16,17]. Thereafter, many Mg-based BMGCs have been developed and all exhibit significant improvements in plasticity as well as toughness. Among these developed Mg-based BMGCs, one special Mg-based BMGC with porous Mo [18,19] performs the optimum combination of yield strength (1100 MPa) and plasticity (>25%), which was chosen as the substrate for further study. Nevertheless, inheriting the reactive characteristics of Mg-based alloys, this Mg-based BMGC is still concerning as it cannot sustain corrosion attacks due to the salty atmosphere of island environments. Therefore, surface treatment for protecting the Mg-based BMGC from the corrosion of a salty atmosphere is essential in the island environment. For conventional Mg alloys, anodic surface treatment [20–22] and micro-arc surface treatment [23,24] are commonly applied to form a protective oxide layer on the Mg alloy surface to prevent the attack of a salty atmosphere in the island environment. However, these two treatments belong to wet processes with alkaline electrolyte solution and have the pollution concern of wastewater. Therefore, a dry process, i.e., sputtering coating treatment [25], is believed to be a better green process for using on the surface treatment of Mg-based alloys. In parallel, Zr-based metallic glasses are reported not only to possess good mechanical properties but also to have excellent corrosion resistance in salty aqueous solutions [26–28]. Hence, coating a thin layer of Zr-based metallic glass thin film (MGTF) on the surface of Mg-based BMGC by sputtering is suggested as an effective and green approach to improve its corrosion resistance in the salty atmosphere.

Accordingly, to further investigate the effect of Zr-based MGTF coating on the corrosion resistance as well mechanical properties of Mg-based BMGC [16], the Zr-based BMG (($Zr_{48}Cu_{36}Al_8Ag_8$)$_{99.5}Si_{0.5}$) was selected as the target material for the sputtering materials due to its high glass-forming ability (GFA), high corrosion resistance, and good mechanical properties. The as-polished specimens of $Mg_{58}Cu_{28.5}Gd_{11}Ag_{2.5}$ BMGC with 25 vol.% Mo particle additions were chosen as the substrate to coat with a 200-nm thick Zr-based MGTF coupled with different thin film buffer layers by the sputtering method. Then, the microstructures, mechanical properties, and corrosion resistances of these MGTF-coated samples were systematically investigated.

## 2. Materials and Methods

### 2.1. Preparation of Sputtering Target

First, we carefully measured pure elements of Zr, Cu, Ni, Al, and Si based on the composition of ($Zr_{53}Cu_{30}Ni_9Al_8$)$_{99.5}Si_{0.5}$. Then, we prepared the ingot via arc-melting 4 times in a Ti-gettered argon atmosphere furnace to assure homogeneity. The final product, a plate with dimensions of 30 mm in *width*, 80 mm in *length* and 8 mm in *thickness*, was produced via casting the ingot into a water-cooling copper mold by suction. The plates were cut with wire-cut Electrical Discharge Machining (EDM) and then assembled into the target with 8 mm in thickness and 3 inches in diameter, as shown in Figure 1. Table 1 shows the chemical composition of the Zr-based target which was firstly examined by EDS before the sputtering process.

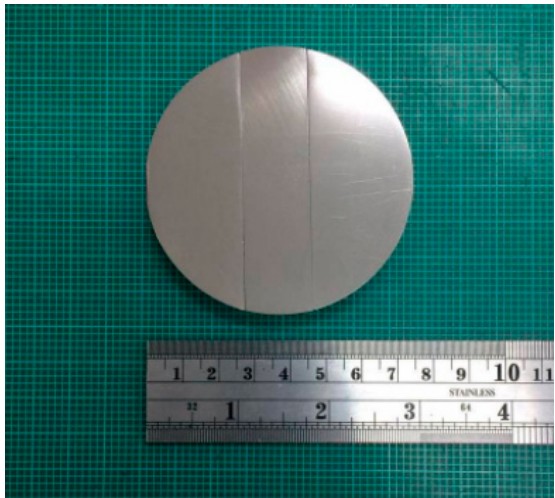

**Figure 1.** The appearance of the assembled Zr-based target with dimensions of 76 mm in *diameter* and 8 mm in *thickness* for the sputtering process.

**Table 1.** Chemical composition of Mg-based bulk metallic glass (BMG) and Mg-based bulk metallic glass composite (BMGC) analyzed by EDS.

| Material | (at.%) | Mg | Cu | Gd |
|---|---|---|---|---|
| Mg-based BMG | Design composition | 58 | 31 | 11 |
| | Average | 51.94 | 34.78 | 13.28 |
| | Deviation | 0.3 | 0.38 | 0.02 |
| Mg-based BMGC | Design composition | 58 | 31 | 11 |
| | Average | 53.7 | 34.09 | 12.22 |
| | Deviation | 0.82 | 0.58 | 0.02 |

## 2.2. Sample Preparation of Mg-Based BMGC

The composition of $Mg_{58}Cu_{28.5}Gd_{11}Ag_{2.5}$ was selected as the base alloy for preparing the BMGC with the addition of 25% porous Mo particles (with average particle size $25 \pm 4$ μm). The composite master alloy ingots were prepared by following the process procedure from our previous report [16]. Then, these composite alloy ingots were further re-melted by induction melting in a quartz tube and injected into a water-cooled Cu mold by argon pressure to obtain the BMGC plates with dimensions of 50 mm $L \times 15$ mm $W \times 3$ mm $T$. The temperature of the Cu mold was kept at 8 °C to reach a cooling rate of 63 K/s for 2-mm thick plates and to obtain a BMGC plate with residual porosity less than 0.15 vol.% [17]. Samples for the three-point bending tests were taken from the as-cast Mg-based BMG and BMGC plates with sample dimensions of 4 mm $W \times 3$ mm $T \times 35$ mm $L$ (of $B$ (thickness) = 2.5 mm, $W$ (width) = 7.5 mm, and $S$ (span) = 36 mm). The as-machined and fine polished BMGC samples were then deposited with two different combinations of thin film coating; Film A: 50-nm thick Cu buffer layer plus 200-nm thick Zr-based MGTF, and Film B: 25-nm thick Al/25-nm thick Ti buffer layer plus 200-nm thick Zr-based MGTF with DC sputtering system (MDX1000, Advanced Energy Industries, Denver, CO, USA). The operating parameters of the DC sputtering procedure were set as follows: the distance between the specimen and target was 10 cm with a base pressure of $10^{-5}$ Pa and working pressure at 0.5 Pa. In parallel, the Ar flow was set at 5.4 sccm, with a sputtering time period of 30 min and 20 W of sputtering power. In addition, an attached test piece for coating thickness examination was coated at the same sputtering conditions as the specimen of the bending test.

### 2.3. Characterization of Microstructure and Properties

The amorphous states of the as-cast Mg-based BMG and BMGC were examined by X-ray diffraction (XRD, Bruker D8A, Cu-Kα radiation, Billerica, MA, USA) and the amorphous state of Zr-based MGTF coating was examined by grazing incident X-ray diffraction analysis (GIXRD, Philips Xpert-Pro PW-3040, Amsterdam, The Netherlands, operated at 40 kV and a 0.5-degree incident angle) with mono-chromatic Cu-Kα radiation. The thickness and composition of the MGTF coating were examined by field emission scanning electron microscopy (FESEM, FEI INSPECT F50, Waltham, MA, USA, operated at 20 kV) with energy dispersion spectroscopy (EDS) at the cross-section of a coupled test specimen. The adhesion capability between the thin film coating and substrate was evaluated by tape testing, which follows the standard of ASTM D3359-09 Test Method B [29]—Cross cut. The hardness of the Mg-based BMG and BMGC was tested by Vickers' hardness tester and that of the Zr-based MGTF coating was checked by a nano-indenter (Hystron, TI 950 Tribo-Indenter, St, Eden Prairie, MN, USA). Following the standard of ASTM E855-08 [30], three-point bending tests were conducted by a universal testing system (MTS Criterion Modle42, Eden Prairie, MN, USA) equipped with a bending gauge, as shown schematically in Figure 2. Before the bending test, the average values of surface roughness of all specimens were confirmed to be less than 10 nm by examination with an atomic force microscope (AFM, Bruker Dimension edge, Billericacity, MA, USA). The morphologies of fractured surfaces after the bending test were examined by FESEM. To further investigate the electrochemical behavior and corrosion properties, 0.9 wt.% NaCl solution was chosen to be the corrosion environment. The Mg-based BMG, Mg-based BMGC, and Mg-based BMGC with Film B were studied by potential dynamic polarization measurements which were conducted by the Autolab PGStat 302 potentiostat (Utrecht, The Netherlands) in a three-electrode cell. The counter and reference electrodes (Saturated Calomel Electrode, SCE) were platinum wire and Ag/AgCl, and specimens with an immersion area of about 25 mm$^2$ were used as a working electrode. The polarization scan was started from −1.5 to 1.5 V with a scan rate 20 mV/s. Corrosion behavior indicators such as corrosion current density ($I_{corr}$), corrosion potential ($E_{corr}$), and corrosion rate can be obtained by the Tafel extrapolation method from anodic polarization curves.

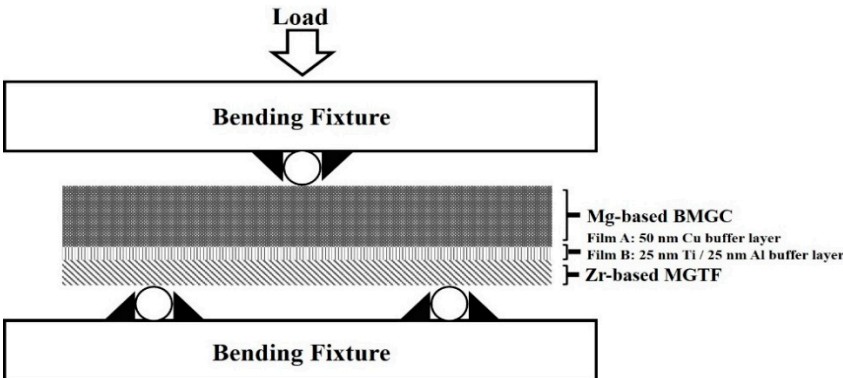

**Figure 2.** Schematic illustration of three-point bending and sample dimension.

## 3. Results

The Zr-based MGTF coatings were firstly examined using EDS to confirm their composition in comparison with the Zr-based MG target. The results of EDS confirmed that the composition of the thin film was close to its pre-set composition, as shown in Table 2. The coating thickness of attached test pieces for different combinations of buffer layer and Zr-based MGTF was found very close to the preset thickness (they are around 50 and 200 nm, respectively), as shown in Figure 3. In addition, the XRD patterns reveal that the Zr-based MGTF coating, the Mg-based BMG, and the Mg-based BMGC all present the amorphous state (typical broadened and diffused humps around 30–50 degrees of 2θ), except the high-intensity crystalline peaks resulting from the Mo particles embedded in the

Mg-based BMGC samples, as shown in Figure 4. In parallel, the average surface roughness can be decreased from 10 (bare Mg-based BMGC substrate) to 4 nm by coating with Zr-based MGTF. This is similar to the results of a published report [31].

**Table 2.** Chemical composition of Zr-based MGTF coating analyzed by EDS.

| Material | (at.%) | Zr | Cu | Ni | Al | Si |
|---|---|---|---|---|---|---|
| Zr-based MGTF | Design composition | 52.73 | 29.85 | 8.96 | 7.96 | 0.5 |
| | Average | 52.44 | 31.69 | 12.05 | 3.69 | 0.13 |
| | Deviation | 0.59 | 0.19 | 0.02 | 0.25 | 0.01 |

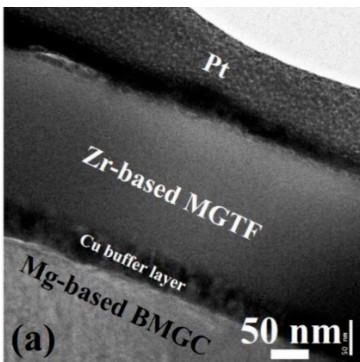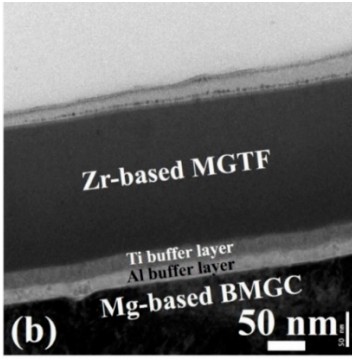

**Figure 3.** Cross-sectional TEM images of the Mg-based BMGC coated with a buffer layer and Zr-based metallic glass thin film (MGTF); (**a**) with Film A coating; (**b**) with Film B coating.

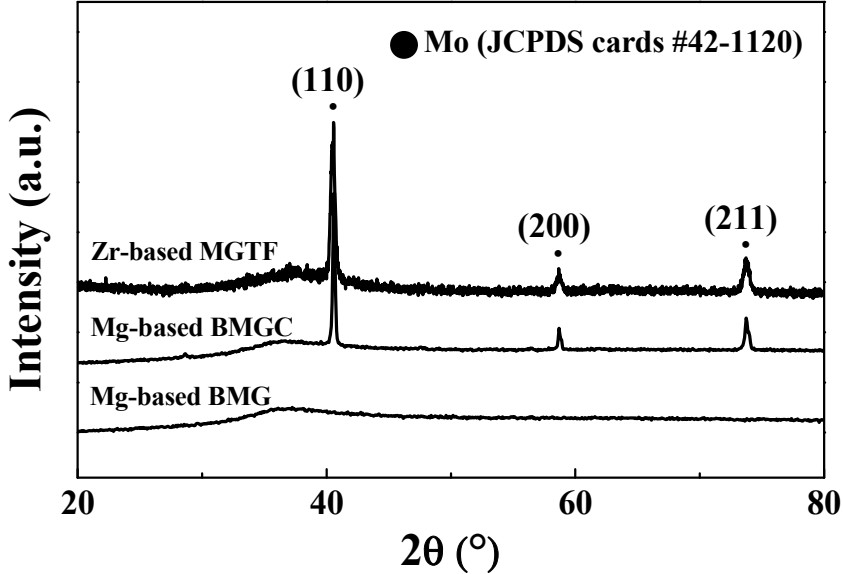

**Figure 4.** X-ray diffraction patterns of Mg-based BMG, Mg-based BMGC, and Mg-based BMGC coated with Zr-based MGTF.

The results of adhesion testing show that the buffer layer coating does affect the adhesion capability of the Zr-based MGTF coatings on the Mg-based BMGC substrate, as shown in Figure 5. The buffer layer of 25-nm thick Al/25-nm thick Ti (Film B) possesses much better adhesion capability than the buffer layer of 50-nm thick Cu (Film A), and only Film B can reach a 4B grade and meet the industrial requirement. This is presumed to be attributed to the large atomic size misfit at the interface of Mg (rMg = 1.60 nm)/Cu (rCu = 1.28 nm) and results in a weak adhesion. On the contrary, the atomic

size misfits at the interfaces of Mg (rMg = 1.60 nm)/Al (rAl = 1.43 nm) and Al (rAl = 1.43 nm)/Ti (rAl = 1.47 nm) are much smaller than the Mg/Cu one.

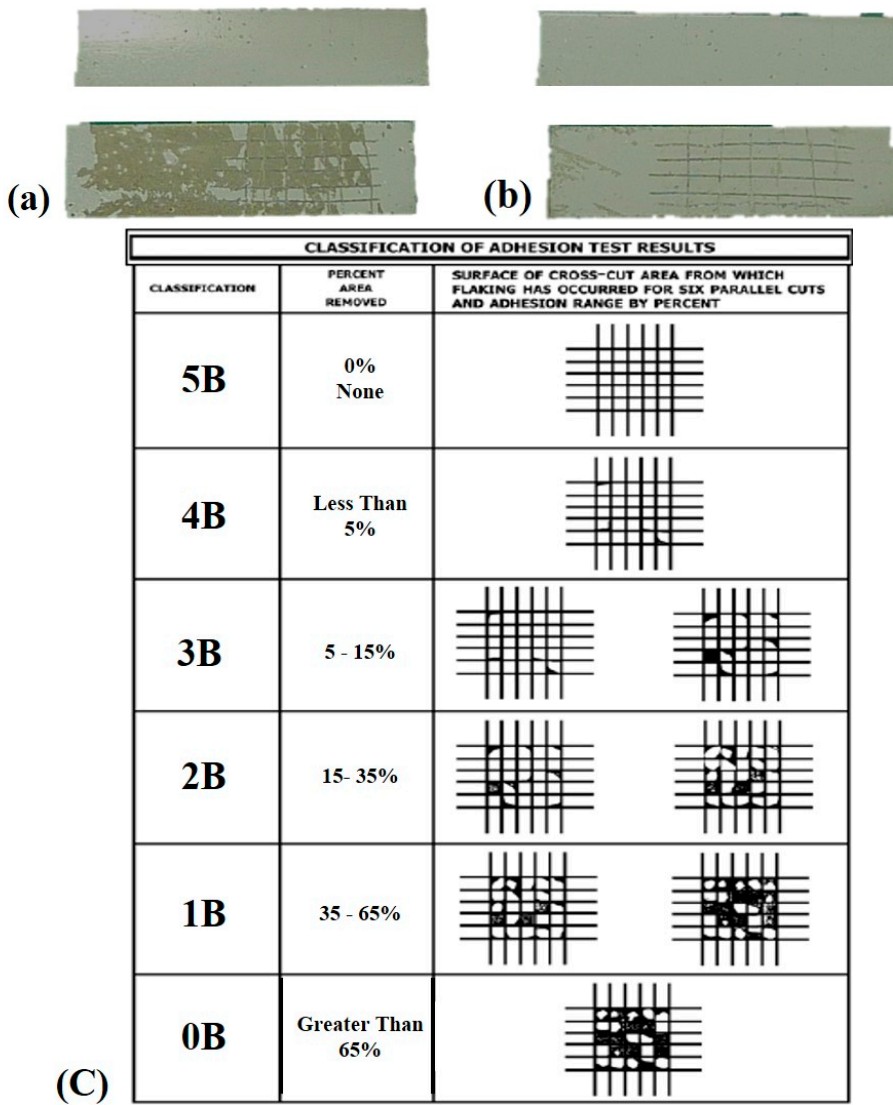

**Figure 5.** Results of adhesion test of the Zr-based MGTF coating on the Mg-based BMGC with different buffer layers; (**a**) with Film A coating; (**b**) with Film B coating. (**c**) Classification of adhesion test results (ASTM D3359-09) [29].

In addition, the results of the bending tests also reveal the significant improvement in the bending fracture strength after coating with Zr-based MGTF, as shown in Figure 6. The bending fracture strength can be improved from 180 (the bare Mg-based BMGC) to 254 MPa (Mg-based BMGC coated with Film B). However, the Mg-based BMGC coated with Film A only presents a slight increase in bending fracture strength (189 MPa) compared with the bare Mg-based BMGC (180 MPa) due to the weak adhesion of the buffer layer between the Zr-based MGTF and Mg-based BMGC.

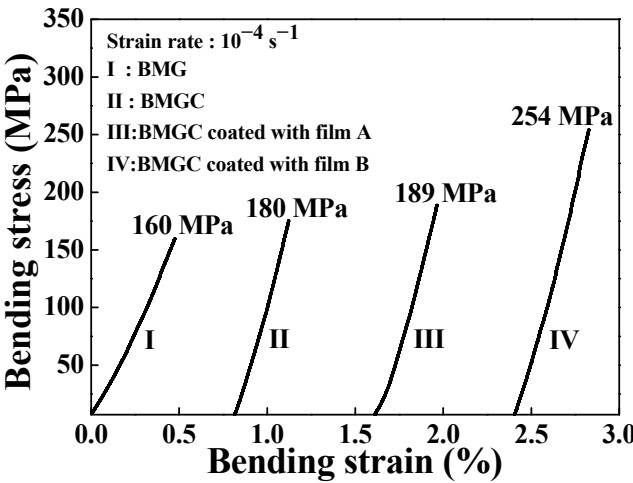

**Figure 6.** Stress-strain curves of bending tests for the samples of Mg-based BMG, Mg-based BMGC, and Mg-based BMGC with Zr-based MGTF coating.

According to the SEM observations, it is revealed that the fracture edge and fracture surface for each sample after bending test exhibit distinct different morphologies, as illustrated in Figure 7. The Mg-based BMG sample only presents very brittle fracture behavior, a sharp fracture edge, and a flat fracture surface. However, the Mg-based BMGC sample shows some cracking traces near the fracture edge with little rough fracture surface, indicating that it has better toughness than the Mg-based BMG, which is in agreement with the previous report [18]. Moreover, the sample of Mg-based BMGC with Film B coating shows a more different morphology near the fracture edge, with many cracks accompanied by several remelting-like traces on the surface of the Zr-based MGTF and relatively rough fracture surface.

In the literature, the improvement in the bending strength and ductility of the MGTF-coated BMG sample has been proposed to be attributed to several major factors [32]: (1) mechanical properties of the thin film coating; (2) surface roughness of the coating; (3) adhesion capability of the coating on the substrate; (4) the flexibility of thin film coating. Accordingly, the significant improvement in the bending strength of the Mg-based BMGC with a Zr-based MGTF coating in this study is suggested to be due to the high strength and good flexibility of the Zr-based MGTF coating [33–35] and the good adhesion of MGTF/Ti/Al buffer-layer coating to substrate [25].

Figure 8 shows the results of the potentiodynamic polarization test in the 0.9 wt.% NaCl solutions of the Mg-based BMG, Mg-based BMGC, and Mg-based BMGC with Zr-based MGTF coating. The free corrosion potential ($E_{corr}$) and the corrosion current density ($I_{corr}$) can be measured from the polarization curves and are listed in Table 3. Values of the corrosion voltage readings for the Mg-based BMG, Mg-based BMGC, and Mg-based BMGC with Zr-based MGTF coating are about −1.04, −1.06, and 0.8 V, respectively. This indicates that the Mg-based BMGC with Zr-based MGTF coating needs to be polarized further before it starts to corrode. In addition, the corrosion current densities for the Mg-based BMG, Mg-based BMGC, and Mg-based BMGC with Zr-based MGTF coating estimated by the Tafel slope method are about $9.07 \times 10^{-5}$, $5.38 \times 10^{-4}$, and $9.06 \times 10^{-6}$ A/cm$^2$, respectively, in 0.9 wt.% NaCl solution, as listed in Table 3. In a comparison of the polarization curves in a 0.9 wt.% NaCl solution amount of the three samples, the Zr-based MGTF coating provided much better corrosion resistance than the bare substrate of Mg-based BMG and BMGC due to a relatively small corrosion current density ($I_{corr}$).

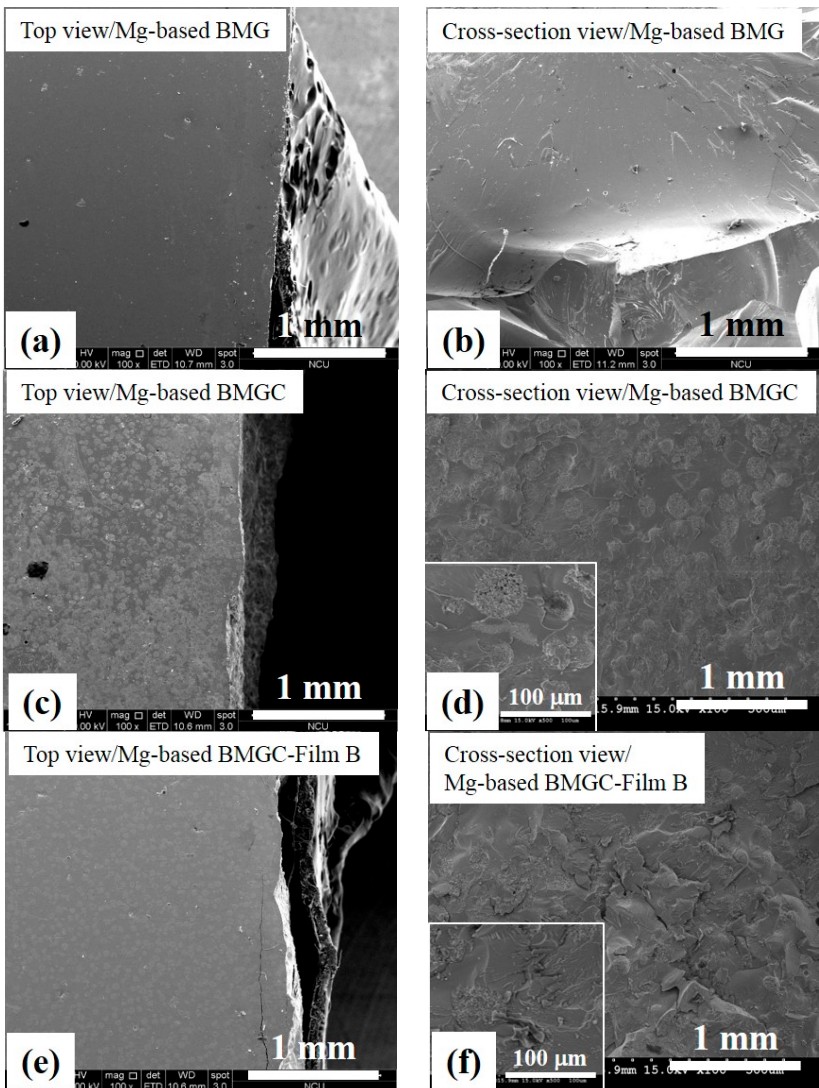

**Figure 7.** SEM images of the fracture edges and fracture surfaces for the samples after bending tests. (**a**) and (**b**) are the Mg-based BMG; (**c**) and (**d**) are the Mg-based BMGC; (**e**) and (**f**) are the Mg-based BMGC with Zr-based MGTF coating. Each insert in (**d**,**f**) is the enlarged image from the circle area of each figure.

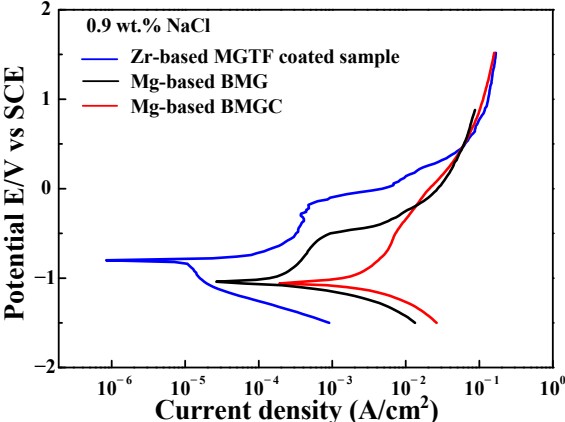

**Figure 8.** Potential dynamic polarization curves of Mg-based BMG, Mg-based BMGC, and Mg-based BMGC with Zr-based MGTF coating in 0.9 wt.% NaCl solution.

**Table 3.** Electrochemical parameters of the polarization test for Mg-based BMG, Mg-based BMGC, and Mg-based BMGC with Zr-based MGTF coating in 0.9 wt.% NaCl solution.

| Material | $E_{corr}$ (V) | $I_{corr}$ (A/cm$^2$) |
|---|---|---|
| Mg-based BMG | −1.04 | $9.07 \times 10^{-5}$ |
| Mg-based BMGC | −1.06 | $5.38 \times 10^{-4}$ |
| Zr-based MGTF | −0.8 | $9.06 \times 10^{-6}$ |

The potential is related to SCE (Ag/AgCl).

## 4. Conclusions

This study revealed that the smooth surface, the excellent adhesion, and the high strength of the Zr-based MGTF have a significant effect on improving the bending strength of Mg-based BMGC. By means of a thin-layer Zr-based MGTF coating accompanied with the Ti/Al buffer-layered coating, the bending strength of Mg-based BMGC could be increased from 180 (bare substrate) to 254 MPa (MGTF-coated sample), which is a 41% improvement. The superior mechanical properties of the Zr-based MGTF such as the high strength and great flexibility, accompanied with the good adhesion to the substrate by Ti/Al buffer-layered coating are the major factors to improve the bending strength of Mg-based BMGC. In addition, the Zr-based MGTF exhibits much better corrosion resistance in 0.9 wt.% sodium chloride solution than the Mg-based BMGC. Therefore, adding a 200-nm thick Zr-based MGTF coating on the Mg-based BMGC by sputtering is believed to be a promising method to protect the Mg-based BMGC from the island environment for many industrial applications.

**Author Contributions:** Conceptualization, J.S.-C.J.; methodology, J.S.-C.J. and P.-H.T.; formal analysis, C.-I.L. and P.-H.T.; investigation, C.-I.L., Y.-C.L., S.-M.S., and T.-H.L.; resources, J.P.C.; data curation, P.-H.T. and T.-H.L.; writing—original draft preparation, P.-H.T.; writing—review and editing, J.S.-C.J.; supervision, J.S.-C.J. All authors have read and agreed to the published version of the manuscript.

**Funding:** This research was funded by the Ministry of Science and Technology of ROC, under the Project MOST 103-2221-E-008 -028 -MY3 and MOST 105-2918-I-008-001.

**Acknowledgments:** The authors gratefully acknowledge the support from the Ministry of Science and Technology of Taiwan, ROC, under the projects MOST 103-2221-E-008-028-MY3 and MOST 105-2918-I-008-001, and the support for analysis from the Precision Instrument Center of National Central University. We also acknowledge the support for analysis from J.P.Chu (National Taiwan University of Science and Technology) and J.W.Lee (Ming-Chi University of Technology).

**Conflicts of Interest:** The authors declare no conflict of interest.

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
