# Peer review of "Improved Mechanical Properties and Corrosion Resistance of Mg-Based Bulk Metallic Glass Composite by Coating with Zr-Based Metallic Glass Thin Film"

_coatings, doi:10.3390/coatings10121212_

Round 1

Reviewer 1 Report

Authors report  on improvement of mechanical properties and corrosion resistance of Mg-based metallic glass composite by 200 nm Zr-based metallic glass thin film application.

I have few minor comments:

  • Line 81: units are missing.
  • Figure captions: please specify film A and B for easier reading.
  • Did the auhors perhaps also compare the surface after corrosion experiment to evaluate the type/degree of corrosion effect?

Overall, the paper is written consistently and I recommend it for publication.

Author Response

Response :

Dear editor and reviewers:

Thank you for your useful comments and suggestions on our manuscript. We have modified the manuscript accordingly, and the point to point reply for the responses and corrections are descried below. All the corrected parts are highlighted by red color, and the detailed corrections are listed below point by point.

  1. Line 81: units are missing.

Response: The missing unit is modified and mark in red.

  1. Figure captions: please specify film A and B for easier reading.

Response: The code name of Film A and Film B were first mentioned at Materiars and Method, paragraph B which illustrates the definition. The Film A is 50 nm-thick Cu buffer layer plus 200 nm-thick Zr-based MGTF and Film B is 25 nm-thick Al/25 nm-thick Ti buffer layer plus 200 nm-thick Zr-based MGTF fabricated via a DC sputtering system, respectively.

  1. Did the auhors perhaps also compare the surface after corrosion experiment to evaluate the type/degree of corrosion effect?

Response: Yes, we did take the surface image after corrosion experiment to verify the behavior of 3 samples. The appearance of the surface corresponds with the polarization test in 0.9 wt.% NaCl solution. Sample with Zr-based MGTF coating is barely damaged in contrast with the Mg-based BMG and BMGC samples. In parallel, We can find many pin holes on the surface of Mg-based BMGC sample which correlate with the porous Mo particle distribution pattern. We speculate that during the casting process the porous Mo particles were not totally filled up with the melt of Mg-based BMG and the left pin hole accelerate the corrode process. Due to lack of strong evidence to support the hypothesis we did not put the result in the manuscript.

Reviewer 2 Report

The article presents the results of the research determining the influence of Zr-based metallic glass thin film on the corrosion resistance and mechanical properties of Mg-based metallic glass composite.

The introduction describes the current state of knowledge sufficiently and justifies the undertaking of research by referring to the literature. However, I believe that the Introduction should include the purpose of the research, but providing a detailed description of the materials here is unnecessary and should be moved to chapter 2.

Moreover, the publications of the co-author "J.S.C. Jang ”are quoted 9 times. For 33 references it gives a significant share of citing own works. It should be verified whether these citations are justified, the more that some of them are not a reference to a single publication but are included in the publication sequence, e.g .:

reference [9] is only quoted in [1-9]

reference [19] is only cited in [14-19]

reference [27] and [28] are only cited in [26-28]

reference [31] is only quoted in [31-33]

The research materials and the research methodology presented were described in detail, but the description of the corrosion test methodology was completely omitted. Information about the electrode with respect to which the potential was determined is given only in Fig. 8, but the abbreviation SCE is nowhere explained.

Figure 2: The orientation of the layer description in Figure 2 makes reading difficult. It should be changed as shown in Fig. 3.

Line 140-143: This content requires reference to the literature because it does not result from the presented research results.

Fig. 5 a and b: The drawings require removing the background and enlarging the samples so that the surface after making the cut is better visible.

Line 186-187. The authors give exact values of the corrosion current, but do not analyze them. Based on the value of the corrosion current, the corrosion rate can be determined.

Author Response

Response:

Dear editor and reviewers:

Thank you for your useful comments and suggestions on our manuscript. We have modified the manuscript accordingly, and the point to point reply for the responses and corrections are descried below. All the corrected parts are highlighted by blue color, and the detailed corrections are listed below point by point.

  1. The introduction describes the current state of knowledge sufficiently and justifies the undertaking of research by referring to the literature. However, I believe that the Introduction should include the purpose of the research, but providing a detailed description of the materials here is unnecessary and should be moved to chapter 2.

Response: The paragraph has been rephrased.

  1. Moreover, the publications of the co-author "J.S.C. Jang ”are quoted 9 times. For 33 references it gives a significant share of citing own works. It should be verified whether these citations are justified, the more that some of them are not a reference to a single publication but are included in the publication sequence, e.g.:

reference [9] is only quoted in [1-9]

reference [19] is only cited in [14-19]

reference [27] and [28] are only cited in [26-28]

reference [31] is only quoted in [31-33]

Response: The paragraph has been rephrased.

  1. The research materials and the research methodology presented were described in detail, but the description of the corrosion test methodology was completely omitted. Information about the electrode with respect to which the potential was determined is given only in Fig. 8, but the abbreviation SCE is nowhere explained.

Response: The description of the corrosion test methodology was added in Materials and Method paragraph C and mark in blue. The three-electrode were also explained in paragraph C and mark in blue.

  1. Figure 2: The orientation of the layer description in Figure 2 makes reading difficult. It should be changed as shown in Fig. 3.

Response: The three-point bending as well as the specimen illustration has been modified.

  1. Line 140-143: This content requires reference to the literature because it does not result from the presented research results.

Response: The information can be find in periodic table and a well-known knowledge.

  1. 5 a and b: The drawings require removing the background and enlarging the samples so that the surface after making the cut is better visible.

Response: We removed the background and enlarged the image for better vision.

  1. Line 186-187. The authors give exact values of the corrosion current, but do not analyze them. Based on the value of the corrosion current, the corrosion rate can be determined.

Response: We need the weight of samples before and after the corrosion experiment for corrosion rate calculation. Due to the time limitation, we apologize that we cannot fabricate samples and do the corrosion experiment again.

Reviewer 3 Report

The paper is well structured, understandable and logic.

However, there are some revisions necessary in order to better support the conclusions and making them comprehensible.

In detail:

Effects shown in figure 7 should be better described within the text and assigned to the parts "a" ... "f". Referring marks, pointing tho the effect described in text, shall be inserted in figure.

Line 181ff:

Please use "free corrosion potential" or "open circuit potential" for these values, since that is the correct term.

Proposal for line 184:

Replace "can sustain a higher voltage before corrosion starts" by "need to be polarized furthermore before it starts to corrode."

Polarization measurements:

How does the specimen's surfaces appear after polarization. Did they really corrode by pitting, or was the current increase caused by other surface reactions? Can you say something on pit depth or repassivation behavior?

Table 3:

Please mention that potential values are related to SCE.

Last conclusion is quite questionable since atmospheric corrosion is different to mechanisms in solutions. At least dry/wet-cycles shall be used for exposure tests.

Author Response

Respond:

Dear editor and reviewers:

Thank you for your useful comments and suggestions on our manuscript. We have modified the manuscript accordingly, and the point to point reply for the responses and corrections are descried below. All the corrected parts are highlighted by green color, and the detailed corrections are listed below point by point.

  1. Effects shown in figure 7 should be better described within the text and assigned to the parts "a" ... "f". Referring marks, pointing to the effect described in text, shall be inserted in figure.

Response: Figure 7 has been modified.

  1. Line 181ff: Please use "free corrosion potential" or "open circuit potential" for these values, since that is the correct term.

Response: Modified.

  1. Proposal for line 184: Replace "can sustain a higher voltage before corrosion starts" by "need to be polarized furthermore before it starts to corrode."

Response: Modified.

  1. Polarization measurements: How does the specimen's surfaces appear after polarization. Did they really corrode by pitting, or was the current increase caused by other surface reactions? Can you say something on pit depth or repassivation behavior?

Response: Yes, we did take the surface image after corrosion experiment to verify the behavior of 3 samples. The appearance of the surface corresponds with the polarization test in 0.9 wt.% NaCl solution. Sample with Zr-based MGTF coating is barely damaged in contrast with the Mg-based BMG and BMGC samples. In parallel, we can find many pin holes on the surface of Mg-based BMGC sample which correlate with the porous Mo particle distribution pattern. We speculate that during the casting process the porous Mo particles were not totally filled up with the melt of Mg-based BMG and the left pin hole accelerate the corrode process. Due to lack of strong evidence to support the hypothesis we did not put the result in the manuscript.

  1. Table 3: Please mention that potential values are related to SCE.

Response: The description of the corrosion test methodology was added in Materials and Method paragraph C.

  1. Last conclusion is quite questionable since atmospheric corrosion is different to mechanisms in solutions. At least dry/wet-cycles shall be used for exposure tests.

Response: Those are previous work; we design only 1-time experiment to verify the anti-corrosion behavior, not the dry/wet-cycles one. Due to the time limitation, we apologize that we cannot fabricate samples and do the corrosion experiment again.

Round 2

Reviewer 2 Report

The article has been significantly improved. The authors responded to all my comments. I believe that the article can be accepted for publication.

Small remarks on the authors' answers.

Answer 4.

In the review, I incorrectly gave fig. 2 for correction, for which I apologize. I meant fig. 3a. The text in this drawing looks like it is upside down. I suggest changing the text direction as shown in red in the attached drawing.

Answer 5, Line 140-144

The presumed causes of weak adhesion are not derived from the periodic table. It would be beneficial to refer to literature here.

Answer 7 Line 186-187

The corrosion resistance can be inferred directly from the value of the corrosion current. No additional research is required here. It would be useful to indicate here that a higher corrosion current indicates that corrosion is occurring at a faster rate.

I leave the presented comments for the authors' consideration.

Author Response

Dear editor and reviewers:

Thank you for your useful comments and suggestions on our manuscript. We have modified the manuscript accordingly, and the point to point reply for the responses and corrections are descried below. Please refer to the change tracing marks of the manuscript, and the detailed corrections are listed below point by point.

  1. Answer 4. In the review, I incorrectly gave fig. 2 for correction, for which I apologize. I meant fig. 3a. The text in this drawing looks like it is upside down. I suggest changing the text direction as shown in red in the attached drawing.

Response: We modified the direction of Fig 3.

  1. Answer 5, Line 140-144. The presumed causes of weak adhesion are not derived from the periodic table. It would be beneficial to refer to literature here.

Response: We need to introduce a buffer layer due to the positive value of heat of mixing (DHmix) between Mg and Zr. The DHmix of Mg-Al, Al-Ti and Ti-Zr are all negative value. This method had been carry out several times in our previous reach to enhance the thin film adhesion ability. Please refer to the following reference: Y.Z. Chang et al., Thin Solid Films 544 (2013) 331–334 and P.H Tsai et al., Thin Solid Films 561 (2014) 28-32.

  1. Answer 7 Line 186-187. The corrosion resistance can be inferred directly from the value of the corrosion current. No additional research is required here. It would be useful to indicate here that a higher corrosion current indicates that corrosion is occurring at a faster rate.

Response: Thank you for your comments.

Reviewer 3 Report

Thanks for incorporating my comments.

However, when mentioning critical pitting potentials in Table 3, proof of pit formation shall be given. Otherwise it ist just a critical corrosion potential.

Pit formation within a coating is also a quality issue.

Without such discussion the potentiodynamic part is incomplete. If there is still a lack of representative examples the whole part should be removed.

Again, in table 3 (either in caption or heading) it shall be mentioned that potential is related to SCE and "2" shall be "²" for square centimeter.

Author Response

Dear editor and reviewers:

Thank you for your useful comments and suggestions on our manuscript. We have modified the manuscript accordingly, and the point to point reply for the responses and corrections are descried below. Please refer to the change tracing marks of the manuscript, and the detailed corrections are listed below point by point.

  1. However, when mentioning critical pitting potentials in Table 3, proof of pit formation shall be given. Otherwise it ist just a critical corrosion potential. Pit formation within a coating is also a quality issue. Without such discussion the potentiodynamic part is incomplete. If there is still a lack of representative examples the whole part should be removed.

Response: Due to lack of further evidence, we decide to remove the result relate to the pitting.

  1. Again, in table 3 (either in caption or heading) it shall be mentioned that potential is related to SCE and "2" shall be "²" for square centimeter.

Response: Modified.
